# Homozygous *YME1L1* mutation causes mitochondriopathy with optic atrophy and mitochondrial network fragmentation

Bianca Hartmann[1,2,3†], Timothy Wai[4†‡], Hao Hu[5,6§], Thomas MacVicar[4], Luciana Musante[5], Björn Fischer-Zirnsak[5,7], Werner Stenzel[8], Ralph Gräf[9], Lambert van den Heuvel[10], Hans-Hilger Ropers[5], Thomas F Wienker[5], Christoph Hübner[2], Thomas Langer[4], Angela M Kaindl[1,2,3*]

[1]Institute of Cell Biology and Neurobiology, Charité University Medicine, Berlin, Germany; [2]Department of Pediatric Neurology, Charité University Medicine, Berlin, Germany; [3]Sozialpädiatrisches Zentrum (SPZ), Center for Chronically Sick Children, Charité University Medicine, Berlin, Germany; [4]Cologne Excellence Cluster on Cellular Stress Responses in Aging-Associated Diseases, Cologne, Germany; [5]Max Planck Institute for Molecular Genetics, Berlin, Germany; [6]Guangzhou Women and Children's Medical Center, Guangzhou, China; [7]Institut of Medical Genetics and Human Genetics, Charité University Medicine, Berlin, Germany; [8]Institute of Neuropathology, Charité University Medicine, Berlin, Germany; [9]Department of Cell Biology, University of Potsdam, Potsdam, Germany; [10]Nijmegen Center for Mitochondrial Disorders, Radboud University Medical Center, Nijmegen, Netherlands

*For correspondence: angela.kaindl@charite.de

†These authors contributed equally to this work

Present address: ‡Institut Necker Enfants Malades, INSERM U1151, CNRS UMR 8253, University Paris Descartes, Paris, France; §Guangzhou Women and Children's Medical Center, Guangzhou, China

Competing interests: The authors declare that no competing interests exist.

**Abstract** Mitochondriopathies often present clinically as multisystemic disorders of primarily high-energy consuming organs. Assembly, turnover, and surveillance of mitochondrial proteins are essential for mitochondrial function and a key task of AAA family members of metalloproteases. We identified a homozygous mutation in the nuclear encoded mitochondrial escape 1-like 1 gene *YME1L1*, member of the AAA protease family, as a cause of a novel mitochondriopathy in a consanguineous pedigree of Saudi Arabian descent. The homozygous missense mutation, located in a highly conserved region in the mitochondrial pre-sequence, inhibits cleavage of YME1L1 by the mitochondrial processing peptidase, which culminates in the rapid degradation of YME1L1 precursor protein. Impaired YME1L1 function causes a proliferation defect and mitochondrial network fragmentation due to abnormal processing of OPA1. Our results identify mutations in *YME1L1* as a cause of a mitochondriopathy with optic nerve atrophy highlighting the importance of YME1L1 for mitochondrial functionality in humans.

## Introduction

Mitochondriopathies often present as multisystemic diseases and can be caused by mutations in nuclear or mitochondrial DNA-encoded genes (*Nunnari, 2012*). Here, we report a novel form of mitochondriopathy with infantile-onset developmental delay, muscle weakness, ataxia, and optic nerve atrophy caused by a homozygous mutation in the yeast mitochondrial escape 1-like 1 gene *YME1L1*.

YME1L1 was first identified in yeast, in a screen for gene products that elevate the rate of mitochondrial DNA migration to the nucleus (*Campbell et al., 1994*; *Thorsness et al., 1993*). The lack of

yme1 impairs respiratory growth of yeast highlighting its important function in mitochondrial maintenance (*Nakai et al., 1995*; *Pearce and Sherman, 1995*). YME1L1, a member of the AAA family of ATPases (ATPases associated with a variety of cell activities), is a nuclear genome-encoded ATP-dependent metalloprotease embedded in the inner mitochondrial membrane (IM), with its protease domain facing the intermembrane space (IMS) (also termed *i*-AAA protease) (*Leonhard et al., 1996*; *Shah et al., 2000*; *Weber et al., 1996*). Its import into mitochondria is accompanied by proteolytic processing via the mitochondrial processing peptidase (MPP), which cleaves off the mitochondrial targeting sequence (MTS) (*Rainey et al., 2006*). The mature protein assembles into a homo-oligomeric complex within the IM (*Leonhard et al., 1996*; *Van Dyck and Langer, 1999*). YME1L1 degrades both IMS and IM proteins such as lipid transfer proteins (*Potting et al., 2013*), components of protein translocases of the IM (*Baker et al., 2012*; *Rainbolt et al., 2015*), and the dynamin-like GTPase optic atrophy 1 (OPA1) (*Anand et al., 2014*; *Rainbolt et al., 2015*; *van der Bliek and Koehler, 2003*).

The *OPA1* gene (MIM*605290), mutated in dominant optic atrophy (*Delettre et al., 2000*), encodes a mediator of mitochondrial fusion that also orchestrates mitochondrial cristae morphogenesis (*Cipolat et al., 2004*; *Frezza et al., 2006*; *Olichon et al., 2003*). OPA1 is processed by two peptidases in the IM, YME1L1 and OMA1 (overlapping with the m-AAA protease 1 homolog), which thereby balance fusion and fission of mitochondria (*MacVicar and Langer, 2016*). Long OPA1 forms (L-OPA1) carry out mitochondrial fusion, while short forms (S-OPA1) are dispensable for fusion but contribute to mitochondrial fission when accumulated (*Anand et al., 2014*; *Ehses et al., 2009*; *Griparic et al., 2007*; *Head et al., 2009*; *Song et al., 2007*). Loss of YME1L1 accelerates OMA1-dependent L-OPA1 cleavage, resulting in S-OPA1 accumulation, increased mitochondrial fission, and mitochondrial network fragmentation (*Anand et al., 2014*; *Wai et al., 2015*). Despite previous attempts to link YME1L1 to human disease (*Coenen et al., 2005*), its physiological role in humans remains to be elucidated.

## Results and discussion

### Homozygous *YME1L1* missense mutation causes mitochondriopathy

We report for the first time that patients with a homozygous mutation in the *YME1L1* gene develop an infantile-onset mitochondriopathy. Four affected children of healthy, consanguineous parents of Saudi Arabian descent were born at term without complications and with normal anthropometric data (*Figure 1A*). Medical history gave no indication for perinatal brain damage as a cause for the reported patient phenotypes. All four patients presented with intellectual disability, motor developmental delay, expressive speech delay, optic nerve atrophy associated with visual impairment, hearing impairment, but no facial dysmorphism (*Figure 1B*, *Table 1*). Inconsistent features were microcephaly (II.9, II.11) or macrocephaly (II.5), ataxia (II.5, II.8, II.9), hyperkinesia (II.9), and athetotic and stereotypic movements (II.11). Cranial MRI revealed leukoencephalopathy in all patients and signs of brain atrophy in patients II.9 and II.11 (*Figure 1C*). Lactate levels and lactate/pyruvate ratios were elevated in blood and/or cerebrospinal fluid (CSF) of three patients as indicators of a mitochondriopathy, while alanine levels were normal in blood and cerebrospinal fluid samples. Analysis of a muscle biopsy specimen from patient II.5 revealed mitochondria with altered cristae morphology (so-called 'parking lots') and paracristalline inclusions as well as a neurogenic pattern with grouped fibers indicating denervation, but no conspicuous ragged red fibers in the Gomori trichrome staining (*Figure 1D*). Results of chromosome analysis, mitochondrial DNA sequencing and genetic testing for fragile X syndrome, Nijmegen breakage syndrome, and ataxia teleangiectasia were normal. Similarly, biochemical analyses gave no indication for adrenoleukodystrophy, GM1/GM2 gangliosidoses, Sandhoff disease, Tay-Sachs disease, Gaucher disease, Fabry disease, Krabbe disease, Mucopolysaccharidosis type IVB, neuronal ceroid-lipofuscinoses 1 and 2, and congenital disorder of glycosylation (CDG) syndrome (data not shown).

To identify the genetic basis of the disease, we performed whole-exome sequencing (WES) followed by Sanger sequencing and bioinformatic analysis. We thereby identified a homozygous missense mutation in a highly conserved region of the *YME1L1* gene in the affected children (c.616C>T, NM_014263; chr10:27425300; GRCh 38.5, raw data available: sequence reads archive (SDA) accession no. SRP073309) that segregates with the phenotype in the index family. We did not detect

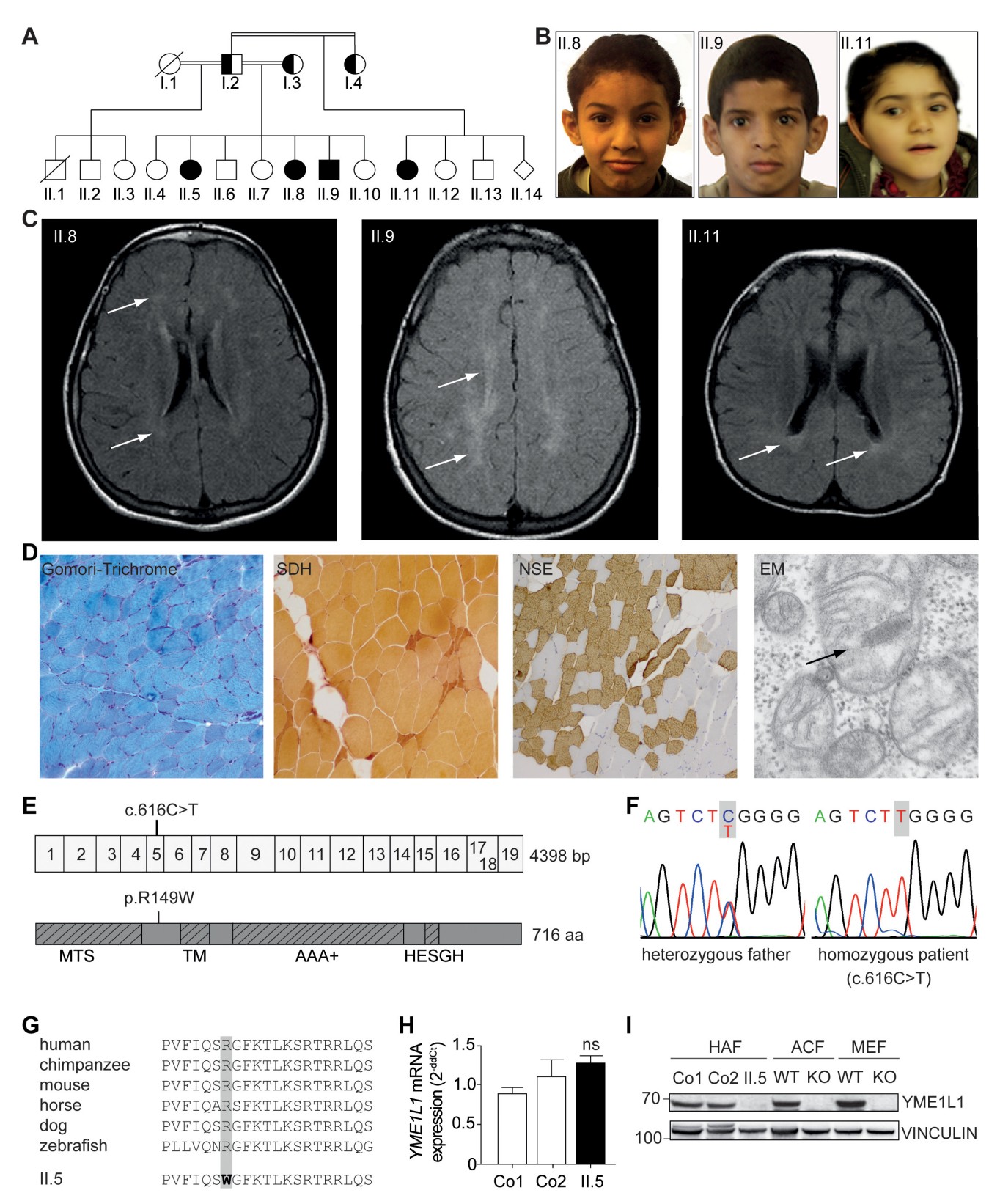

**Figure 1.** Phenotype and genotype of patients with YME1L1 mitochondriopathy. (A, B) The four affected patients are children of healthy, consanguineous parents of Saudi Arabian decent (□ male; ○ female; ◊ unknown gender; ⌀⌀ deceased; ◨◖ heterozygous, clinically not affected; ●/■

*Figure 1 continued on next page*

*Figure 1 continued*

homozygous, affected; ⬚, consanguineous marriage). (**C**) Cranial MRIs reveal hyperintense changes (arrows) as a sign for leucencephalopathy, and cerebral atrophy (FLAIR, axial images). (**D**) Histological analysis of patient II.5 muscle biopsy specimen, from left to right: Gomori trichrome stain for muscle fibers revealed no conspicuous ragged red fibers indicative of a mitochondriopathy. Succinate dehydrogenase (SDH) and neuron specific enolase (NSE) staining revealed a neurogenic pattern with grouped fibers indicating denervation (magnification 200x). Electron microscopy (EM) revealed paracristalline inclusions (arrow) and altered cristae structure ("parking lots", magnification 15,000x). (**E**) Whole exome sequencing discovered the homozygous mutation c.616C>T in exon 5 of the *YME1L1* gene (NM_014263), localized at position 149 of the YME1L1 protein, and leading to an amino acid exchange of arginine to tryptophan in the mitochondrial targeting site (MTS; p.R149W, NP_055078). YME1L1 contains highly conserved domains: MTS, transmembrane domain (TM), ATPase domain (AAA+), motif of metalloprotease (Zinc) activity (HESGH). (**F**) Electropherogram depicting homozygous missense mutation in patient II.5, which is heterozygous in the father. (**G**) The mutation lies within a protein region, highly conserved throughout different species. (**H**) YME1L1 mRNA levels do not differ between patient and control primary human adult fibroblasts (Co, Control; II.5, patient II.5; ns, not significant; one-way ANOVA; p=0.2314; n=6 ) (**I**) Steady state levels of YME1L1 are below detection levels or profoundly reduced in cell lysates of patient II.5 YME1L1$^{R149W}$; *Yme1l1* knockout mouse fibroblasts serve as negative controls. YME1L1 protein levels in mitchondrial compartment fractions can be found *Figure 1—figure supplement 1* (HAF, primary human adult fibroblasts; ACF, immortalized murine adult cardiac fibroblasts; MEF, immortalized murine embryonic fibroblast; Co, Control; II.5, patient II.5; WT, wild type; KO, *Yme1l1* knockout, n=5).

The following source data and figure supplement are available for figure 1:

**Source data 1.** Raw data Graph 1H.

**Figure supplement 1.** (I) YME1L1$^{R149W}$ signal is barely detectable in total cell lysates (T), significantly reduced in the mitochondrial (M) but not present in the cytosolic (C) fraction of patient primary human adult fibroblasts (Co, Control; II.5, patient II.5; SDHA, succinate dehydrogenase; LDHA, lactate dehydrogenase; n=3).

mutations in other genes linked previously to neurologic diseases. (*De Michele et al., 1998*; *Hazan et al., 1994*; *Warnecke et al., 2007*). The *YME1L1* gene mutation c.616C>T causes the exchange of highly conserved hydrophilic arginine to hydrophobic tryptophan (p.R149W, NP_055078) within the N-terminal MTS (*Figure 1E–G*). YME1L1$^{R149W}$ was below detection levels or profoundly reduced in patient whole cell lysates and in mitochondrial enriched fractions of patient fibroblasts, while mRNA levels remain unaltered (*Figure 1H,I*, *Figure 1–source data 1*, *Figure 1–figure supplement 1*). These results identify a disease-causing mutation in arginine 149 of YME1L1, which impairs protein accumulation.

## YME1L1$^{R149W}$ abrogates maturation of YME1L1 upon import into mitochondria

The affected amino acid residue arginine 149 is present within a predicted mitochondrial targeting sequence of YME1L1 and may affect targeting of the mutant protein to mitochondria. We therefore transiently expressed YME1L1 and YME1L1$^{R149W}$ in *YME1L1*$^{-/-}$ HeLa cells generated by CRISPR/Cas9-mediated genome editing (*Figure 2A*). Both YME1L1 and YME1L1$^{R149W}$ co-localized with the mitochondrial marker protein ATPase subunit ß, demonstrating mitochondrial targeting of the mutant protein. Consistently, inhibition of the proteasome did not stabilize mutant YME1L1 (*Figure 2B*), suggesting that reduced levels of YME1L1$^{R149W}$ are not caused by proteasomal degradation of non-imported YME1L1 precursor proteins. We therefore excluded a degradation of mutant YME1L1$^{R149W}$ via the proteasome.

Nuclear-encoded YME1L1 requires cleavage of the MTS from premature YME1L1 (~80 kDa) upon import into mitochondria to give rise to the mature protein (~63 kDa), which is able to assemble into a homooligomeric proteolytic complex in the IM (*Graef et al., 2007*; *Stiburek et al., 2012*; *Van Dyck and Langer, 1999*). MTS cleavage of YME1L1 precursor is bioinformatically predicted to be mediated by MPP (NM_014263; Mitoprot) (*Claros and Vincens, 1996*). Since the identified human mutation affects an arginine and such residues are known to function as MPP recognition and cleavage sites (*Niidome et al., 1994*), we asked whether arginine 149 is part of the MPP cleavage site in YME1L1. We mutated various arginine residues in the N-terminal region of YME1L1 and analyzed maturation of corresponding YME1L1 variants synthesized in a cell-free expression system using recombinant MPP (*Figure 2C*). YME1L1 was converted by MPP in its mature form (*Figure 2C*). Mutation of arginine at position 149 abrogated MPP-mediated processing of YME1L1, suggesting

**Table 1.** *YME1L1* mitochondriopathy phenotype.

| Pedigree ID (gender) | | | II.5 (f) | II.8 (f) | II.9 (m) | II.11 (f) |
|---|---|---|---|---|---|---|
| Age at last assessment (y) | | | 15.8 | 12.3 | 10.3 | 5.2 |
| Category | Feature | HPO | | | | |
| *Inheritance* | | | | | | |
| | | | AR | AR | AR | AR |
| *Growth* | | | | | | |
| Height | Short stature<br>SD;% | 0004322 | +<br>-2.19; 1 | +<br>-2.09; 2 | +<br>-1.89; 3 | -<br>0.38; 35 |
| Weight | Low weight<br>SD;% | 0004325 | +<br>-2.8; <1 | -<br>-1.18; 12 | -<br>0.07; 53 | -<br>-1.35; 12 |
| *Neonatal period* | | | | | | |
| | Neonatal asphyxia | 0012768 | - | - | - | + |
| *Head and Neck* | | | | | | |
| Head | Microcephaly | 0000252 | - | - | + | + |
| | Macrocephaly | 0000256 | + | - | - | - |
| Face | Midface retrusion | 0011800 | - | - | + | + |
| | Congenital facial diplegia | 0007188 | + | - | - | + |
| Ears | Hearing impairment | 0000365 | - | + | + | + |
| | Sensorineural hearing impairment | 0000407 | n.a. | n.a. | + | + |
| | Macrotia | 0000400 | - | + | + | - |
| Eyes | Pigmentary retinopathy | 0000580 | + | - | - | - |
| | Optic nerve hypoplasia | 0000609 | + | + | + | + |
| | Cherry red spot of the macula | 0010729 | - | - | - | + |
| | Strabismus | 0000486 | - | - | + | + |
| | Hypermetropia | 0000540 | - | - | - | + |
| | Myopia | 0000545 | + | + | + | - |
| | Amblyopia | 0000646 | - | + | + | + |
| | Abnormality of visual evoked potentials | 0000649 | n.a. | - | + | + |
| *Abdomen* | | | | | | |
| Gastro-intestinal | Constipation (in infancy) | 0002019 | + | n.a. | + | + |
| Spleen | Splenomegaly | 0001744 | + | - | + | - |
| *Skeletal* | | | | | | |
| Feet | Bilateral talipes equinovarus | 0001776 | - | - | - | + |
| *Muscle, soft tissue* | | | | | | |
| | Increased variability in muscle fiber diameter | 0003557 | + | n.a. | n.a. | n.a. |
| *Neurologic* | | | | | | |
| CNS | Hypotonia, neonatal, generalized | 0008935 | - | - | - | + |
| | Infantile muscular hypotonia | 0008947 | - | - | - | + |
| | Global developmental delay (onset, months) | 0001263 | + | + | + | + (6) |
| | Motor delay | 0001270 | + | + | + | + |
| | Gait apraxia | 0010521 | - | - | - | + |
| | Athetosis | 0002305 | - | - | - | + |
| | Intellectual disability, moderate; (IQ) | 0002342 | + (48) | n.a. | n.a. | + (39) |
| | Incomprehensible speech | 0002546 | n.a. | - | - | + |
| | Poor speech | 0002465 | n.a. | - | + | - |
| | Absent speech | 0001344 | - | + | - | - |

*Table 1 continued on next page*

Hartmann *et al.* eLife 2016;5:e16078. DOI: 10.7554/eLife.16078

*Table 1 continued*

| Pedigree ID (gender) | | | II.5 (f) | II.8 (f) | II.9 (m) | II.11 (f) |
|---|---|---|---|---|---|---|
| | Seizures (onset, months) | 0001250 | - | - | - | + (6) |
| | Dysmetria | 0001310 | + | + | + | - |
| | Ataxia | 0001251 | + | + | + | - |
| | Brain atrophy | 0012444 | - | - | - | + |
| | Ventriculomegaly | 0002119 | - | - | - | + |
| | Delayed CNS myelination | 0002188 | + | - | - | - |
| | Abnormality of the cerebral white matter | 0002500 | + | + | + | + |
| | Abnormality of the basal ganglia | 0002134 | - | - | + | - |
| | Cerebellar hypoplasia (progressive) | 0001321 | + | - | - | + |
| | EEG with focal sharp waves | 0011196 | n.a. | - | + | - |
| | Abnormal auditory evoked potentials | 0006958 | n.a. | n.a. | + | + |
| PNS | Decreased sensory nerve conduction velocity | 0003448 | - | + | n.a. | - |
| Behavioural Psychiatric | Hyperactivity | 0000752 | - | + | - | - |
| | Attention deficit hyperactivity disorder | 0007018 | - | - | - | + |
| | Stereotypical body rocking | 0012172 | - | - | - | + |
| *Laboratory anomalies* | | | | | | |
| | Mildly elevated creatine phosphokinase | 0008180 | - | - | - | + |

that amino acid 151 forms the N-terminal amino acid of mature YME1L1. Replacement of arginine 149 by tryptophan similarly abolished maturation of YME1L1 by MPP in vitro (*Figure 2D*).

## YME1L1$^{R149W}$ undergoes rapid proteolysis

Grossly reduced YME1L1 protein levels coinciding with unaltered mRNA levels indicate a post-transcriptional mechanism responsible for the reduction of YME1L1$^{R149W}$. Stabilization of the precursor form of YME1L1 by abolishing the MPP recognition site may allow another peptidase to cleave YME1L1. To identify the latter mechanism, we screened candidate mitochondrial proteases for their effect on YME1L1 and YME1L1$^{R149W}$ stably expressed in HEK293T cells, but could not detect a difference in YME1L1/YME1L1$^{R149W}$ level upon depletion of individual proteases (data not shown). However, by preventing the proteolytic activity of YME1L1 upon ectopic expression of the dominant-negative YME1L1$^{E381Q}$, harboring a point mutation in the ATPase domain, we were able to stabilize not only the known YME1L1 substrate PRELID1 (*Potting et al., 2013*) but also observed accumulation of both the precursor and mature forms of YME1L1 (*Figure 3A*). The mature form of YME1L1 remained stable in cells harboring YME1L1$^{E381Q}$ upon further incubation in the presence of cycloheximide, while ectopic expression of the wild type variant resulted in significantly lower levels of YME1L1 and reduced stability (*Figure 3A*). These observations point to an auto-catalytic degradation of YME1L1$^{R149W}$. We therefore combined the dominant negative mutation E381Q with the pathogenic mutation R149W in YME1L1 in cis and examined the stability of this variant (*Figure 3A*). We observed an accumulation of mature YME1L1$^{R149W/E381Q}$ upon ectopic expression in HEK293T cells (*Figure 3A*). Both mature YME1L1$^{R149W/E381Q}$ and larger forms with a similar size to the precursor form of YME1L1 accumulated in these cells. Stabilization of the precursor form of YME1L1 in these cells may allow another peptidase to cleave YME1L1 resulting in the accumulation of slightly smaller forms of YME1L1. These results suggest that YME1L1 mediates degradation of mutant YME1L1$^{R149W}$, which accumulates in the precursor form due to the mutation of the MPP cleavage site and thus is likely recognized as a misfolded protein in the IM. As the mutant YME1L1 variants were expressed in wild type HEK293T cells, we cannot distinguish whether degradation of YME1L1$^{R149W}$ occurs intramolecularly and/or by endogenous YME1L1 under these experimental conditions. However, the lack of wild-type YME1L1 in patient cells with homozygous *YME1L1* mutations together with our ectopic expression studies argue for auto-catalytic degradation of YME1L1$^{R149W}$.

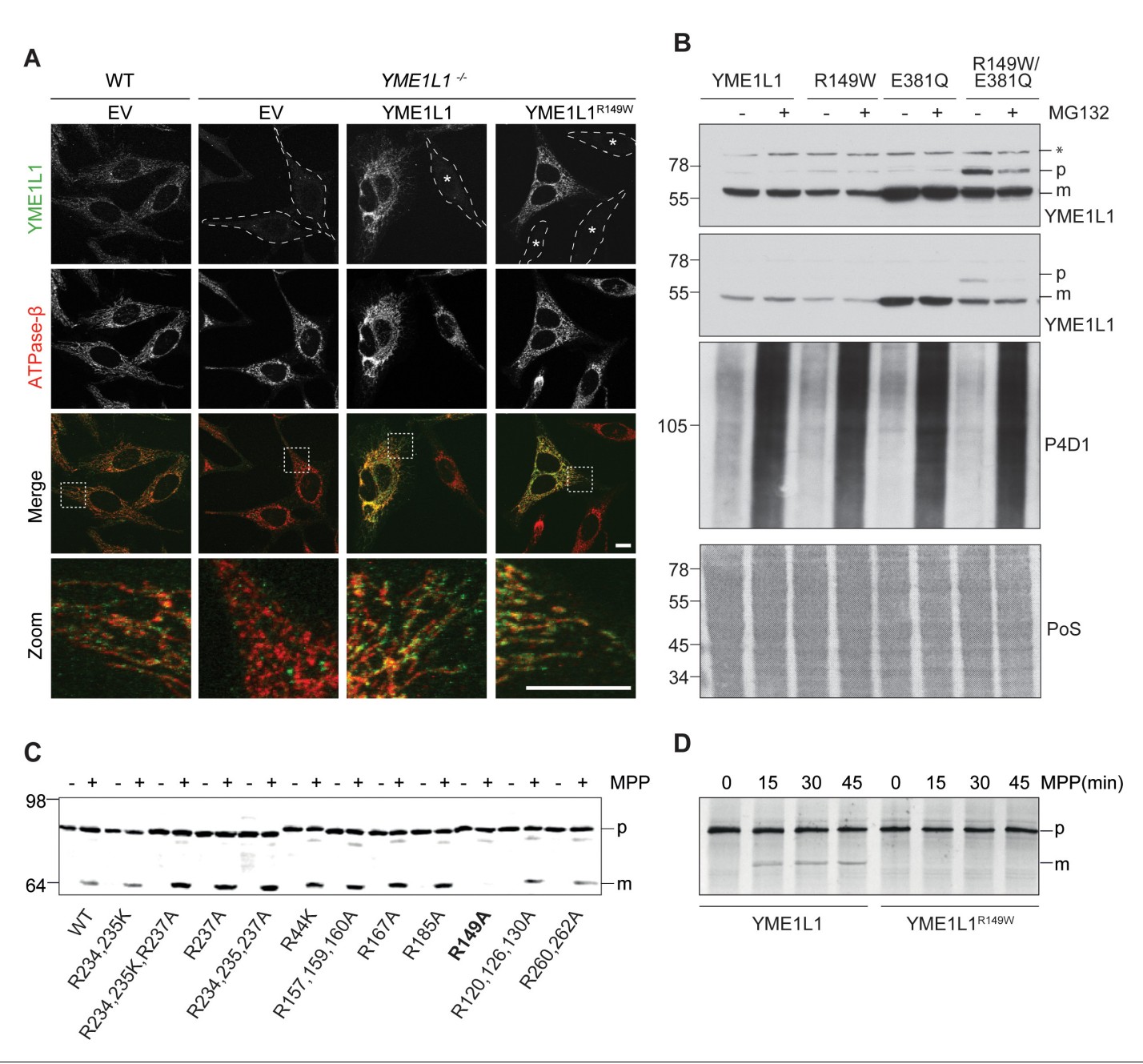

**Figure 2.** Mutation of arginine 149 impairs maturation of YME1L1 by MPP upon import into mitochondria. (**A**) YME1L1[R149W] is targeted to mitochondria. HeLa cells were transiently transfected with wild-type YME1L1, YME1L1[R149W] or the empty vector (EV) as control. Cells were analyzed by indirect immunofluorescence and co-localization of the immunofluorescent signals of antibodies against YME1L1 and ATP Synthase subunit beta (ATPase-ß) (WT, Wildtype; YME1L1[-/-], Knockout; EV, empty vector; YME1L1[R149W], patient mutation; Scale bar 10 µM). (**B**) Inhibition of the proteasome with MG132 (20 mM, 18 h) does not stabilize precursor or mature YME1L1 in Flp-In T-Rex HEK293T cells expressing YME1L1 or YME1L1 mutant variants: R149W, E381Q or R149W/E381Q. P4D1 antibodies were used as a control for ubiquitin accumulation and proteasome inhibition. (R149W, patient mutation; E381Q, dominant negative mutation of ATPase domain; R149W/E381Q, double mutant; *, unspecific signal; p, premature; m, mature; PoS, Ponceau staining; n=2). (**C**) Maturation of YME1L1 is mediated by mitochondrial processing peptidase (MPP) and impaired upon mutation of arginine 149. After site-directed mutagenesis of N-terminal arginine residues in YME1L1, the mutant proteins were expressed in a cell-free system and processing by recombinant MPP was examined (WT, wild type YME1L1; p, premature m, mature, n=2–3). (**D**) Cell-free MPP cleavage-assay: MPP can cleave YME1L1 but not YME1L1[R149W] from the premature to its mature form (min, minutes; p, premature; m, mature; YME1L1[R149W], patient mutation; n=3).

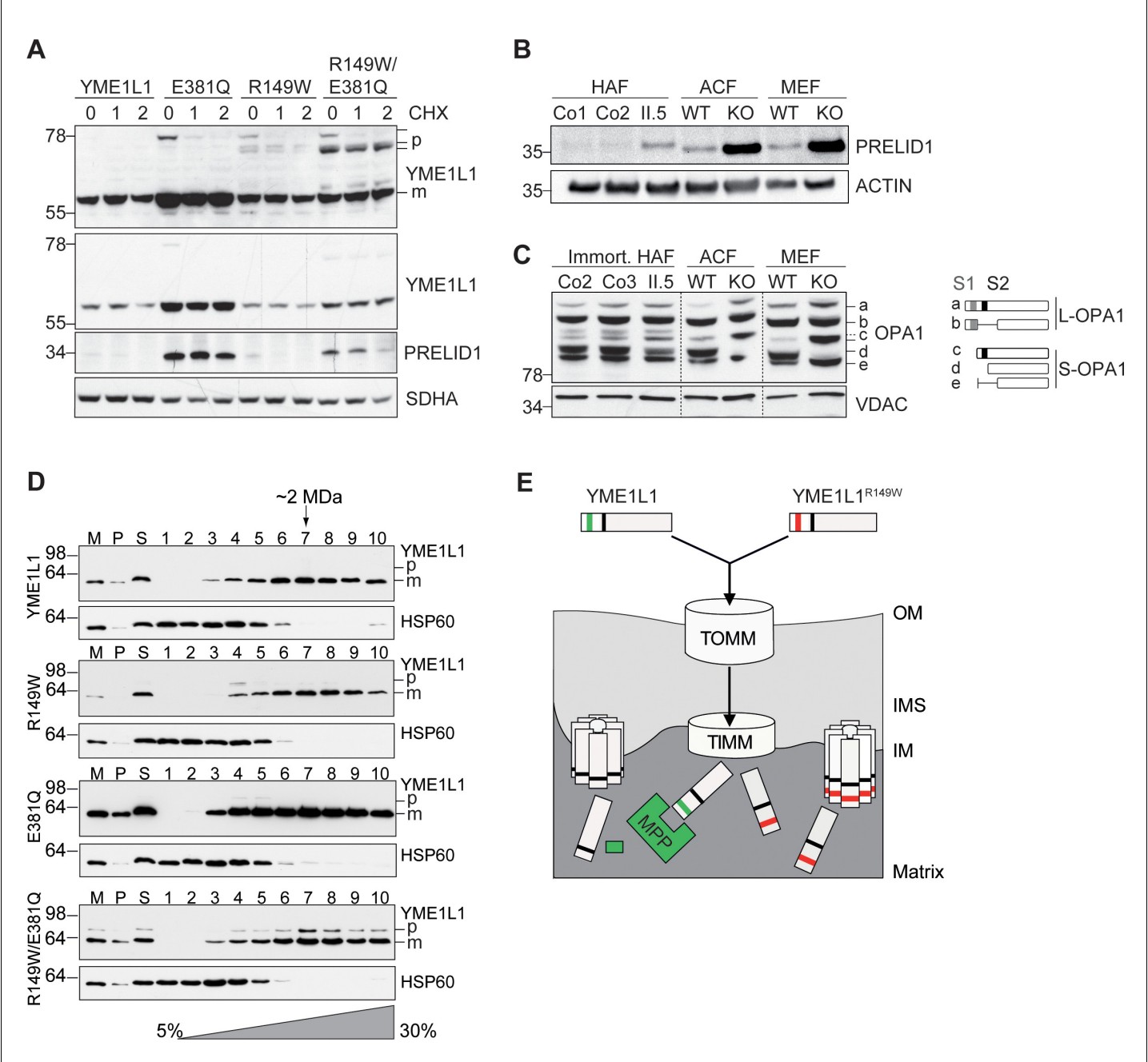

**Figure 3.** Mutation of arginine 149 destabilizes YME1L1 but retains residual YME1L activity. (**A**) Stability of YME1L1 or YME1L1 mutant variants (R149W, E381Q and R149W/E381Q) expressed in Flp-In T-Rex HEK293T cells. The dominant negative E381Q mutation in the ATPase domain of YME1L1 prevents degradation of YME1L1$^{R149W}$ (E381Q, dominant negative mutation of ATPase domain; R149W, patient mutation; R149W/E381Q, double mutant; CHX, cycloheximide; h, hours; p, premature; m, mature; SDHA, succinate dehydrogenase; n=2). (**B**) Homozygous mutation in *YME1L1* results in an accumulation of PRELID1 in the human patient. *Yme1l1* knockout mouse fibroblasts serve as positive controls to demonstrate impaired proteolysis of PRELID1 (HAF, human adult primary fibroblasts; ACF, immortalized murine adult cardiac fibroblasts; MEF, immortalized murine embryonic fibroblast; Co, Control; II.5, patient; WT, wild type; KO, knockout; n=5). (**C**) Mutation of arginine 149 of YME1L1 impairs processing of OPA1 with a decrease of short OPA1 form d levels. The formation of OPA1 form d indicates residual YME1L1 activity in human patient fibroblasts (immort. HAF, immortalized human adult fibroblasts; ACF, immortalized murine adult cardiac fibroblasts; MEF, immortalized murine embryonic fibroblast; Co, Control; II.5, patient; WT, wild type; KO, knockout; n=6). The schematic diagram illustrates the proteolytic processing of OPA1 by YME1L1 on processing site 2 (S2) and OMA1 on processing site 1 (S1). The presence of long OPA1 forms (L-OPA1) is required for the maintenance of mitochondrial inner membrane fusion, whereas accumulation of short OPA1 forms (S-OPA1) is associated with accelerated fission. (**D**) Mitochondria-enriched membrane fractions from Flp-In T-Rex HEK293T cells expressing YME1L1 or YME1L1 mutant variants (R149W, E381Q and R149W/E381Q) were solubilized in digitonin and

*Figure 3 continued on next page*

*Figure 3 continued*
analyzed by sucrose gradient centrifugation. Fractions were collected and separated on SDS-PAGE for immunoblotting to detect high MW complexes of YME1L1. HSP60 complexes were used as a control (M, mitochondrial input, P, S, pellet and supernatant fraction after solubilization; HSP60, heat shock protein 60; E381Q, dominant negative mutation of ATPase domain; R149W, patient mutation; R149W/E381Q, double mutant). (E) Premature YME1L1/ YME1L1$^{R149W}$ is imported into the mitochondrial matrix via translocons of the outer mitochondrial membrane (TOMM) and inner mitochondrial membrane (TIMM). Here, MPP binds and cleaves the N-terminal mitochondrial targeting site (MTS) from premature YME1L1 but not YME1L1$^{R149W}$, which then allows the mature and premature YME1L1$^{R149W}$ protein to assemble as proteolytic complex.

## Impaired proteolysis by YME1L1$^{R149W}$

To confirm a functional impairment of YME1L1$^{R149W}$, we analyzed the effect of YME1L1$^{R149W}$ on two substrate proteins, the protein of relevant evolutionary and lymphoid interest (PRELID1) and OPA1, in patient-derived skin fibroblasts. PRELID1, a lipid transfer protein for phosphatidic acid in the IMS, is constitutively degraded by YME1L1 (*Potting et al., 2013*). PRELID1 accumulated in primary human fibroblasts (HAF) of patient II.5 when compared to controls (*Figure 3B*), consistent with the low levels of YME1L1$^{R149W}$ present in mitochondria (*Figure 1I*, *Figure 1–figure supplement 1*). Deletion of murine *Yme1l1* leads to accelerated processing of L-OPA1 by OMA1 (*Anand et al., 2014*; *Rainbolt et al., 2016*) and drives mitochondrial fragmentation in vitro (*Baker et al., 2014*) and in vivo (*Korwitz et al., 2016*; *Wai et al., 2015*). We similarly identified an altered cleavage pattern of OPA1 in patient cells, albeit more modestly than in knockout mouse cells (*Figure 3C*).

This suggests that YME1L1$^{R149W}$ retains proteolytic activity. As proteolysis by YME1L1 depends on its oligomerization, we examined in further experiments the assembly of YME1L1$^{R149W}$ expressed in HEK293 cells by sucrose gradient centrifugation (*Figure 3D*). To allow detection of YME1L1$^{R149W}$ precursor forms and impair its degradation, we also examined the assembly of YME1L1$^{R149W/E381Q}$ expressed in HEK293 cells. Similar to YME1L1, the mutant variants of YME1L1 were recovered in protein complexes of ~2 MDa (*Figure 3D*). Notably, we also observed assembly of the precursor form of YME1L1$^{R149W/E381Q}$, demonstrating that the disease-causing mutation R149W does not impair the formation of the *i*-AAA protease complex. Thus, assembled YME1L1$^{R149W}$ precursor forms may explain the residual activity of YME1L1 in patient fibroblasts (*Figure 3E*). Alternatively, and not mutually exclusive, low levels of mature YME1L1$^{R149W}$ may maintain residual proteolytic activity in the patient system.

Together, we conclude that YME1L1$^{R149W}$ is a hypomorphic allele that retains partial YME1L1 activity, which reconciles the embryonic lethality (*Wai et al., 2015*) of *Yme1l1* knockout mice with the milder phenotype in our patients with a homozygous *YME1L1* missense mutation.

## YME1L1$^{R149W}$ causes mitochondrial fragmentation

Given the role of unbalanced mitochondrial dynamics in neurodegenerative diseases (*Burté et al., 2015*) and the observed, impaired OPA1 processing, we focused our attention on mitochondrial morphology in patient fibroblasts. We detected an increase in shortened and fragmented mitochondrial networks in HAF of patient II.5 relative to a control (*Figure 4A,B*, *Videos 1–4*, *Figure 4–source data 1*), consistent with the more severe fragmentation phenotype in *Yme1l1* KO mouse fibroblasts (*Anand et al., 2014*). We therefore transiently expressed human YME1L1$^{R149W}$ and YME1L1 in murine adult cardiac *Yme1l1$^{-/-}$*fibroblasts (*Wai et al., 2015*) and observed that human YME1L1 but not YME1L1$^{R149W}$ was able to complement mitochondrial fragmentation. Of note, the human mutant variant was still able to partially tubulate mitochondria (*Figure 4C,D*, *Figure 4–source data 1*), consistent with the notion that YME1L1$^{R149W}$ retains residual function.

## YME1L1$^{R149W}$ impairs cell proliferation

To further address the pathophysiological mechanism underlying the neurological phenotype in humans, we analyzed proliferation and apoptosis of HAF. Proliferation of patient II.5 fibroblasts was significantly reduced compared to those of healthy controls, while the level of induced apoptotic cell death remained unaffected (*Figure 5A–C*, *Figure 5–source data 1*, *Figure 5–source data 1*, *Figure 5–source data 1*). Analysis of the respiratory chain in patient fibroblasts revealed the specific enzymatic activities to be within the normal physiological range and subunit levels to be unaltered (*Figure 5D,E*), consistent with the observation in Yme1l1–depleted hearts (*Wai et al., 2015*).

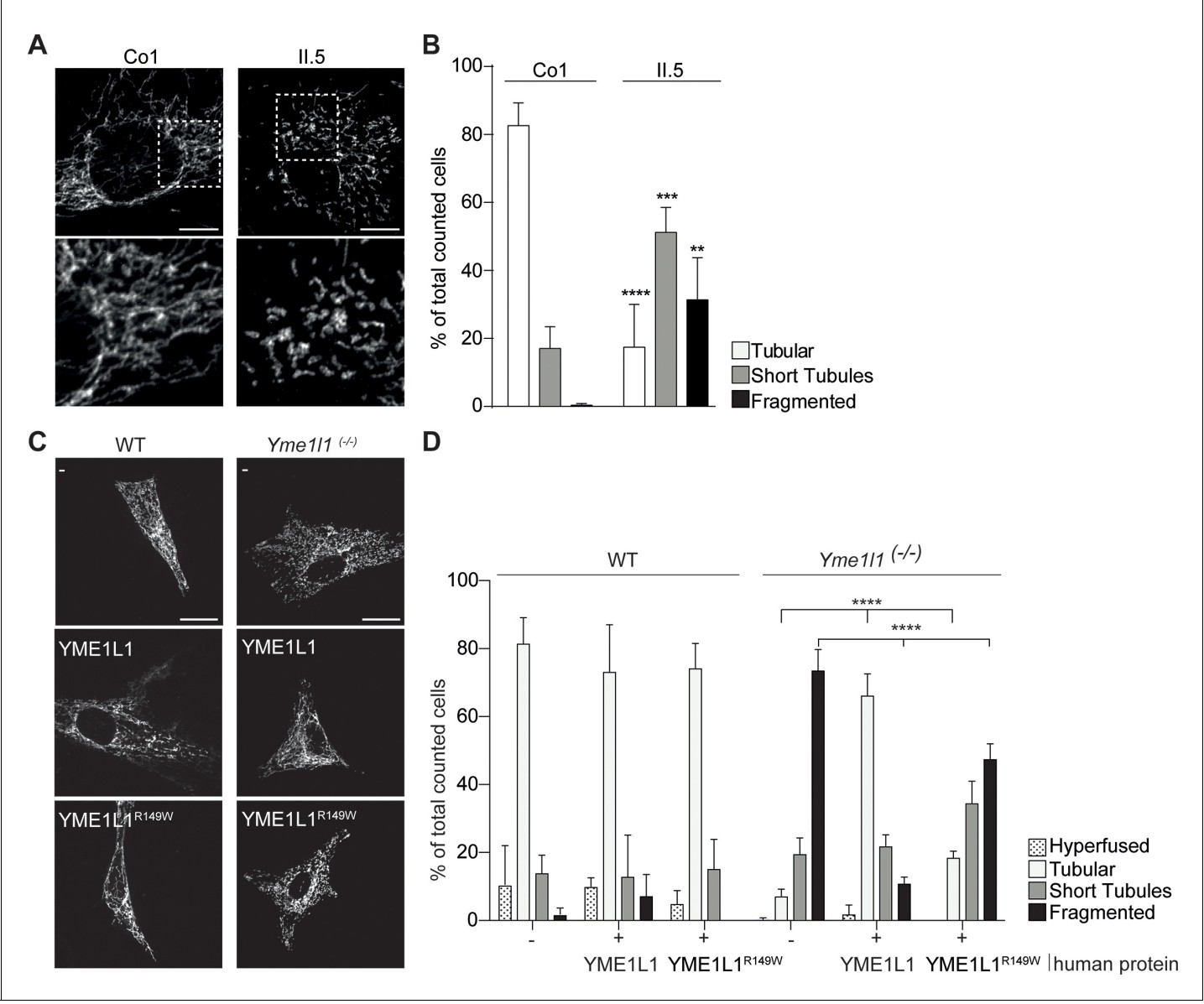

**Figure 4.** YME1L1[R149W] causes mitochondrial fragmentation. (A, B) YME1L1[R149W] causes fragmentation (fragmentation or short tubules) of mitochondrial networks in patient fibroblasts (Co, Control, II.5, patient II.5; one-way ANOVA; n=150–200 cells; 4 replicates; scale bar 10 µm). (C) Co-transfection of mitochondrial GFP and human YME1L1/ YME1L1[R149W] protein in WT and Yme1l1[-/-] MEFs revealed that YME1L1 but not YME1L1[R149W] can rescue mitochondrial fragmentation in Yme1l1[-/-] MEFs and (D) increases the number of cells with tubular networks. Expression of YME1L1[R149W] partially rescues the fragmentation phenotype. In WT MEFs, YME1L1[R149W] expression results in only a mild, but not significant decline of cells with a tubular mitochondrial network (WT, wildtype; Yme1l1[-/-], Yme1l1 Knockout; two-way ANOVA; n=3; scale bar 10 µm; **p<0.01; ***p<0.001; ****p<0.0001).

The following source data is available for figure 4:

**Source data 1.** Raw data Graph 4B.

**Source data 2.** Raw data Graph 4D.

However, the reduced steady state levels of various mitochondrial compartment markers in patient cells indicate a general reduction of mitochondrial mass in YME1L1[R149W] cells (*Figure 5E*, *Figure 5F*).

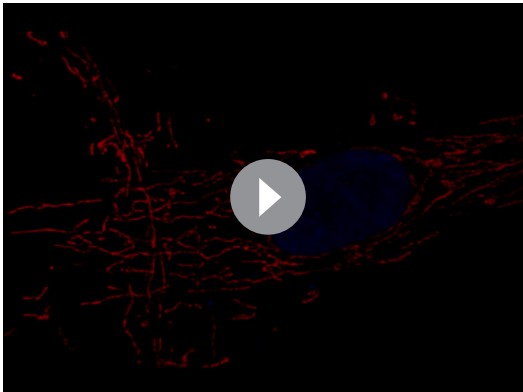

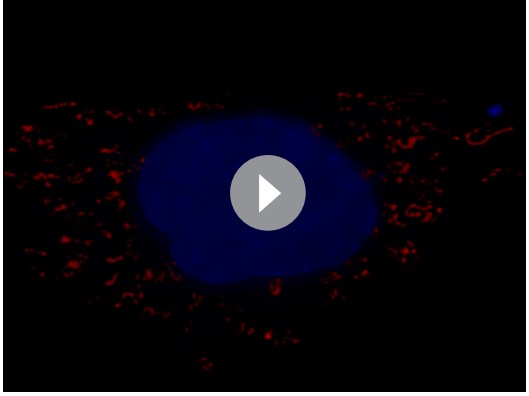

**Video 1.** Three dimensional rendering of mitochondrial structure of a control fibroblast. MitoTracker Deep Red-mitochondria; DAPI-nucleus.

**Video 2.** Three dimensional rendering of mitochondrial structure of a patient II.5 fibroblast. MitoTracker Deep Red- mitochondria; DAPI-nucleus.

Here we report a new form of infantile-onset mitochondriopathy caused by a homozygous mutation in the *YME1L1* gene. Patients suffer from severe intellectual disability, muscular impairments, and optic nerve atrophy. In contrast to complete gene deletion in the mouse (*Wai et al., 2015*), the homozygous c.616C<T missense mutation is compatible with life, likely owing to the residual function of YME1L1^R149W. We demonstrate that the missense mutation affects the MPP processing site and impairs YME1L1 maturation, leading to its rapid degradation. We further show that the YME1L1^R149W leads to a proliferation defect, abnormal OPA1 processing and mitochondrial fragmentation. Since fusion and fission of mitochondria are essential to preserve cellular functions, it is conceivable that defects in these processes, contribute to the patient phenotype in a cell-type specific manner. With our findings, we reveal an important role of YME1L1 for the maintenance of mitochondrial morphology in humans and link a loss of YME1L1 function to disease. YME1L1 thereby joins other members of the AAA family of ATPases that have already been linked to neurologic disease, such as spastin (*SPG4*, MIM*6042779 and paraplegin (*SPG7*, MIM*602783) gene, and in the gene encoding the YME1L1 substrate optic atrophy 1 (*OPA1*, MIM*605290) (*De Michele et al., 1998*; *Hazan et al., 1994*; *Nielsen et al., 1997*; *Warnecke et al., 2007*). Future studies on further patients are required to determine the full phenotype spectrum associated with *YME1L1* gene mutations.

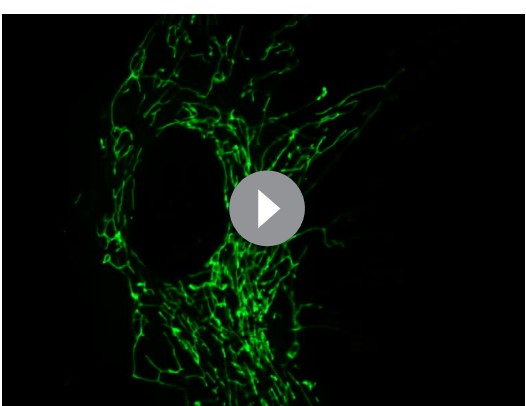

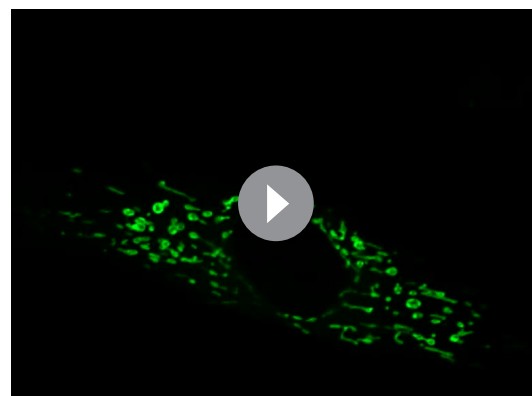

**Video 3.** Mitochondrial dynamics in a control fibroblast. MitoTracker Green-mitochondria, live cell imaging.

**Video 4.** Mitochondrial dynamics in a patient II.5 fibroblast. MitoTracker Green-mitochondria, live cell imaging.

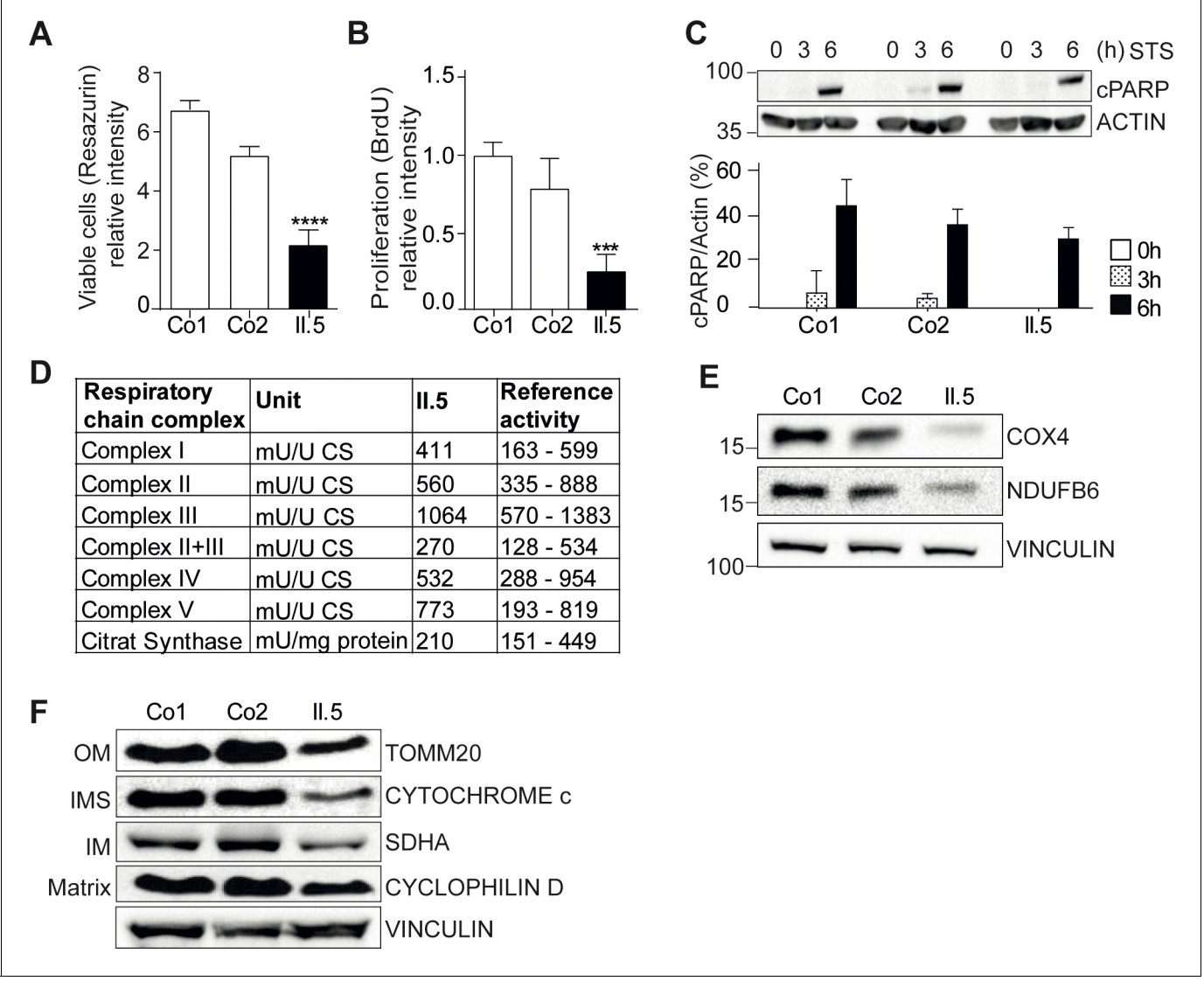

**Figure 5.** YME1L1 mutation impairs fibroblast cell growth and proliferation. (A) Significant reduction of cell culture growth (y-axis scale *$10^4$) and (B) proliferation of patient primary human adult fibroblasts after 120h under culture conditions (Co, Control; II.5, patient II.5; one-way ANOVA; ***p<0.001; ****p<0.0001; n=5 ). (C) No increase in apoptosis-sensitivity upon 2 μM staurosporine (STS) treatment in patient HAF after 6h (SDS-PAGE for cleaved PARP level (cPARP) (Co, Control; II.5, patient II.5; one-way ANOVA; p6h = 0.4283; n=3; ). (D) Normal respiratory chain subunit activity in patient HAF (U, units; mU, milli U; CS, Citrate synthase). (E) Cytochrome c oxidase subunit 4 (COX4) and NADH ubiquinone oxidoreductase 1 beta subcomplex 6 (NDUFB6) protein levels were not elevated in patient fibroblasts (Co, Control; II.5, patient II.5; n=3). (F) Decreased levels of mitochondrial compartment markers TOMM20 (translocon of outer mitochondrial membrane 20) for the outer membrane (OM), Cytochrome c for the intermembrane space (IMS), SDHA (succinate dehydrogenase) for the inner membrane (IM), and Cyclophilin D for the matrix in whole cell lysates of patient II.5 (Co, Control; II.5, patient II.5; n=3).

The following source data is available for figure 5:

**Source data 1.** Raw data Graph 5A.
**Source data 2.** Raw data Graph 5B.
**Source data 3.** Raw data Graph 5C.

## Material and methods

### Subjects

Informed consent was obtained from the parents of the patients for the molecular genetic analysis, the publication of clinical data, photos, magnetic resonance images (MRI), and studies on fibroblasts. The human study was approved by the local ethics committee of the Charité (approval no. EA1/212/08).

### Genetic analyses

Homozygosity linkage intervals (chr10:24,333,063–49,978,774, LOD=2.533) with LOD>2 and length >1 Mb were identified using the Affymetrix SNP array 6.0. For whole exome sequencing, 1 µg genomic DNA extracted from peripheral blood was enriched by Agilent SureSelect Human All Exon enrichment kit version 3 and deep-sequenced by Illumina Hiseq2000 sequencer in a 101 bp single-end mode. The output sequences amounted to 12 Gb and >94% of the exon-coding regions were covered by at least 20 folds. The data analysis pipeline considered linkage interval, pathogenicity of variants, allele frequency in polymorphism databases (1000Genome, ESP, and an in-house database including 868 individual exomes of Middle East origin), functional involvement of specific genes, thus resulting in the identification of a homozygous *YME1L1* gene mutation (chr10p12.1) (*Najmabadi et al., 2011*). We confirmed the absence of homozygous deleterious variants (missense, nonsense, frameshift, splice site change) in *YME1L1* in the NHLBI Exome Sequencing Project (ESP, http://evs.gs.washington.edu/EVS/; ExAC 0.3 http://exac.broadinstitute.org/), the 1000 Genome database (http://www.1000genomes.org/), the dbSNP138 (http://www.ncbi.nlm.nih.gov/SNP/), and our in-house 898 Middle East origin exome database. The raw data is available under the sequence reads archive (SDA) accession no. SRP073309.

Sanger sequencing of the *YME1L1* gene (NM_014263, GRCh 38.5) confirmed the mutation in the patients and established the genotype in the other family members and in additional families.

### Mouse and human fibroblasts

HAF were established from skin biopsies derived from patient II.5 and unrelated controls (Co1-3). MEF and adult cardiac fibroblasts (ACF) were derived from *Yme1l1* mutant mice (*Yme1l1*^loxP/loxP^) as described previously (*Anand et al., 2014*; *Wai et al., 2015*; *Yao and Shoubridge, 1999*). HAF, MEF, and ACF were cultured in high glucose Dulbecco's modified Eagle's medium (Gibco, Darmstadt, Germany) supplemented with 15%, 10%, and 10% fetal bovine serum (FBS, Biochrome, Berlin, Germany; Life Technologies; Carlsbad, California), respectively, and 1% penicillin-streptomycin (Sigma Aldrich; St. Louis, Missouri). Fibroblasts were immortalized by stable expression of hTERT and E7 (*Zhu et al., 1998*).

### Generation of *YME1L1*^-/-^ HeLa cells

*YME1L1* KO cells were generated using the CRISPR/Cas9 system. Guide RNA (gRNA) and Cas9 were expressed using the px330 expression vector (*Cong et al., 2013*). The following gRNA target sequence was used: 5'-GGAACCGACCATATTACAACAGG-3´ (recommended by the lab of Alexander van der Bliek). Single clones were obtained by serial dilution after transfection and screened by SDS-PAGE and immunoblotting.

### qPCR

DNA extraction and cDNA synthesis were performed with established methods reported previously (*Issa et al., 2013*). To specifically amplify and detect murine *Yme1l1*, human *YME1L1*, *Hprt* (Hypoxanthine-guanine phosphoribosyl-transferase, reference gene), and *RPII* (RNA polymerase II, reference gene) cDNA, we designed sets of primers and TaqMan probes specified in *Supplementary file 1A* using the GenScript real-time PCR (TaqMan) Primer Design online software (www.genscript.com). qPCR and quantification was performed as described previously (*Issa et al., 2013*), and further statistics were performed using the GraphPad Prism 5 Software (GraphPad Software Inc., La Jolla, California).

## Western blot

Protein whole lysate extraction, fractionated mitochondrial and cytosolic protein extraction, and Western blots were performed with established methods reported previously (*Issa et al., 2013*; *Potting et al., 2013*). Antibodies are listed in *Supplementary file 1B*.

## Isolation of a crude mitochondrial fraction

The protocol was adapted from (*Vogel et al., 2005*). Cells were harvested using trypsin and washed three times in PBS to remove the trypsin. Subsequently, the cells were resuspended in 3 ml isotonic Buffer (0,25 M Sucrose, 5 mM Tris-HCL pH 7,5, 1 mM EDTA and 0,1 mM PMSF). The cells were opened with eight strokes by 2500 rpm using a potter homogenisator. Unbroken cells and nuclei were removed by centrifugation at 600x g for 15 min at 4°C. The supernatant was applied to another centrifugation step at 10000x g for 25 min at 4°C. The supernatant contains the cytoplasmic fraction (C) whereas mitochondria are enriched in the pellet. The mitochondrial fraction (M) was washed two times in isotonic buffer and afterwards applied to SDS Page.

## Cell viability, apoptosis, and proliferation

Viability and proliferation of HAF were quantified using the Fluorimetric CellTiter-Blue Cell Viability Assay (Promega, Madison, Wisconsin) and the colorimetric Cell Proliferation BrdU-ELISA (Roche Diagnostics, Rotkreuz, Switzerland), respectively, according to the manufacturer´s instructions. Apoptosis of HAF following treatment with staurosporine (Cell signaling, Cambridge, United Kingdom) was determined through quantification of cleaved PARP protein levels on Western blots.

## Mitochondrial respiratory chain enzyme activity

Mitochondrial respiratory chain enzyme activities were measured in skin fibroblasts according to established procedures (*Janssen et al., 2006*; *Smeitink et al., 2001*). The values were expressed relative to the activity of Cytochrome C oxidase and/or Citrate synthase.

## Overexpression of human *YME1L1*

Human wild type *YME1L1* and mutant *YME1L1$^{R149W}$* were stably expressed in Flp-In T-Rex HEK293T cells (Life Technologies) under the control of a tetracycline-regulated promotor according to manufacturer´s instructions. HEK293T cells were cultured as described previously (*Potting et al., 2013*). Transient transfection was performed using GeneJuice (EMD Millipore, Darmstadt, Germany) according to manufacturer´s instruction.

## Sucrose gradient density centrifugation

Cells were harvested and washed three times in PBS. Subsequently, the cells were resuspended in isotonic buffer (0,25 M Sucrose, 5 mM Tris-HCL pH 7,5, 1 mM EDTA and 1X cOmplete, EDTA-free Protease Inhibitor Cocktail [Sigma]). The cells were opened with 15 strokes by 1000 rpm using a Potter S homogenisator (Sartorius, Göttingen, Germany). Unbroken cells and nuclei were removed by centrifugation at 600x g for 5 min at 4°C. The supernatant was applied to another centrifugation step at 10000x g for 10 min at 4°C. The mitochondrial-enriched membrane fraction (pellet) was resuspended in lysis buffer (270 mM sucrose, 100 mM KCl, 20 mM 20 mM $MgCl_2$, 10 mM Tris/HCl, pH 7.5, and 1X cOmplete, EDTA-free Protease Inhibitor Cocktail) containing 6 g of digitonin per g of protein at a concentration of 2.5 mg/ml for 20 min on ice. Supernatant from a clarifying centrifugation at 16,000 g for 10 min at 4°C was directly applied to a 5–25% sucrose gradient prepared using an automated gradient maker (Biocomp instruments, Fredericton, Canada) in 14x89mm Ultra-Clear centrifuge tubes (Beckman Instruments, Brea, California). Ultracentrifugation was performed at 71,000 x g for 16 hr at 4°C and 1.2 ml fractions (n=10) were collected by hand, precipitated by TCA and subjected to SDS-PAGE.

## Cell-free synthesis of YME1L1 and processing by purified MPP in vitro

The cell-free synthesis of (*Baker et al., 2014*) S-labeled precursor proteins of human wild type and mutant YME1L1 was performed using the TNT Sp6 or Coupled Reticulocyte lysate system (Promega) according to the manufacturer's instructions. Radiolabeled wild type and mutant YME1L1 were incubated in cleavage buffer (20 mM HEPES, pH 7,4, 50 mM $NaCl_2$, 1 mM Zn $Cl_2$, 1 mM ATP, 5 mM Mg

Cl$_2$) at 30°C in the presence of absence of MPP as described previously (*Nolden et al., 2005*). Samples were analyzed by SDS-PAGE and autoradiography.

## Immunocytochemistry

Immunocytochemistry was performed as described previously (*Anand et al., 2014*; *Issa et al., 2013*). Antibodies are listed in *Supplementary file 1B*; DAPI was purchased from Sigma-Aldrich.

## Mitochondrial network analysis

HAF, MEF, and ACF were stained with anti-TOMM20 antibody and DAPI (4′,6-Diamidin-2-phenylindol). Images were taken as described below and quantified in double-blind fashion into four categories: hyperfused, tubular, short tubules and fragmented mitochondrial network as previously described (*Anand et al., 2014*). For the complementation experiments: co-transfection of 100 ng of mitochondrially-targeted GFP (mito-GFP) along with 1 µg of either YME1L1 or YME1L1$^{R149W}$ was performed, and GFP-positive fixed cells were scored for mitochondrial morphology.

## Imaging

Fluorescently labeled fibroblasts were imaged with a fluorescent Olympus BX51 microscope using the software Magnafire 2.1B Version 2001 (Olympus, Tokyo, Japan) and a Spinning Disc Microscopy system (Carl Zeiss Microscopy, Oberkochen, Germany) with the ZEN 2012 Software, with an lsm5exciter Zeiss confocal microscope (Carl Zeiss Microscopy). Analysis of overexpression experiment of human wildtype and mutant *YME1L1* was realized with an UltraVIEW VoX spinning disc Yokogawa CSU-X1 confocal microscope (PerkinElmer, Waltham, Massachusetts) on a Nikon Ti microscope equipped with a 60x (Apo TIRF 60x Oil N.A. 1.49) objective and camera (EMCCD C9100-50 Cam-Link). Imaging of HeLa cells was performed using a Meta 510 confocal laser scanning microscope (Carl Zeiss). All images were processed using Adobe Photoshop (Adobe Systems Inc., San José, California), Volocity (PerkinElmer), and ImageJ (National Institute of Health, USA).

## Live-cell imaging

HAF were cultured on 35 mm Fluorodish Cell Culture Dishes (World Precision Instruments, Sarasota, Florida) and incubated in pre-warmed medium containing 500 nM MitoTracker Green (Thermo Fisher Scientific, Waltham, Massachusetts) 30 min prior to imaging. Live-cell imaging was performed in fresh medium in the microscope incubation chamber (5% CO$_2$, 37°C) of a Zeiss Axio Observer with Spinning Disc Technology Carl Zeiss Microscopy) for 7–10 min. Recordings were processed using Adobe Photoshop (Adobe Systems Inc.) and Magix Video Maker Deluxe (Magix Software GmbH, Berlin, Germany).

## Electron microscopy (EM)

Fibroblasts and muscle biopsy specimen were prepared for TEM analysis and imaged as described previously (*Anand et al., 2014*; *Stenzel et al., 2015*; *Wakabayashi et al., 2009*).

## Statistical methods

All results are presented as mean ± SD, GraphPad Prism 5 software was used for all statistical analyses, and the significance level was set at $p < 0.05$. Differences between two groups were evaluated by un-paired or paired Student's t-test, while for multi-group comparisons, one-way ANOVA with Tukey's multiple-comparison test was used. For the mitochondrial fragmentation analysis (*Figure 3D*), two-way ANOVA followed by the Tukey's multiple-comparisons test was used. Sample sizes, replicate numbers, and p-values are stated in the figure legends. Replicates are biological replicates except for experiments illustrated in *Figures 1H* and *2B*.

## Acknowledgements

The authors thank the family members who participated in this study. We acknowledge T. Burmester for sending us fibroblasts of the affected children, E. Morava-Kozicz, D. Emmerich for sending DNA samples of patients with YME1L1 mitochondriopathy-like phenotypes, as well as G. Stoltenburg-Didinger, V. Tarabykin, C. Birchmeier, N. Krämer, K. Voigt, and J. Fassbender for technical help and

discussions. This work was supported by the German Research Foundation (SFB665, KO2891/1-1, Reinhart Koselleck grant), the Berlin Institute of Health (BIH, CRG1), the Sonnenfeld Stiftung, the German Society for Muscle diseases (DGM), the German Academic Exchange Service (DAAD), the NCRR P20-RR016453, the Robert C. Perry Fund (20061479), the Max-Planck Society, the EU FP 7 project GENCODYS (241995), and the Human Frontiers Science Program.

## Additional information

### Funding

| Funder | Grant reference number | Author |
|---|---|---|
| Sonnenfeld Stiftung | | Bianca Hartmann |
| Deutsche Gesellschaft für Muskelkranke e.V. | | Bianca Hartmann |
| Human Frontier Science Program | | Timothy Wai |
| Max-Planck-Gesellschaft | | Hao Hu<br>Luciana Musante<br>Hans-Hilger Ropers<br>Thomas F Wienker |
| Deutsche Forschungsgemeinschaft | KO2891/1-1 | Björn Fischer-Zirnsak |
| National Center for Research Resources | P20 RR016453 | Hans-Hilger Ropers |
| EU FP 7 project GENCODYS | 241995 | Hans-Hilger Ropers |
| Deutsche Forschungsgemeinschaft | Reinhart Koselleck Grant | Thomas Langer |
| Deutsche Forschungsgemeinschaft | SFB665 | Angela M Kaindl |
| Berlin Institute of Health | BIH, CRG1 | Angela M Kaindl |
| Deutscher Akademischer Austauschdienst | | Angela M Kaindl |

The funders had no role in study design, data collection and interpretation, or the decision to submit the work for publication.

### Author contributions

BH, TW, Conception and design, Performed functional experiments, Analysis and interpretation of data, Drafted the manuscript; HH, LM, Generated and analyzed WES and performed further mutation analyses in the cohort, Drafting or revising the article; TM, BF-Z, Performed functional experiments, Analysis and interpretation of data, Drafting or revising the article; WS, Acquisition of data, Aanalyzed patient muscle specimen, Drafting or revising the article; RG, Helped with imaging, Acquisition of data, Analysis and interpretation of data, Drafting or revising the article; LvdH, Performed mitochondrial enzyme studies and related DNA studies, Analysis and interpretation of data, Drafting or revising the article; H-HR, TFW, Conception and design, Generated and analyzed WES and performed further mutation analyses in the cohort, Drafting or revising the article; CH, AMK, Conception and design, Recruited subjects, Gathered patient history as well as clinical information, Contributed clinical samples, Drafting or revising the article; TL, Conception and design, Acquisition of data, Analysis and interpretation of data, Drafted the article

### Author ORCIDs

Angela M Kaindl, http://orcid.org/0000-0001-9454-206X

### Ethics

Human subjects: Informed consent was obtained from the parents of the patients for the molecular genetic analysis, the publication of clinical data, photos, magnetic resonance images (MRI), and

studies on fibroblasts. The human study was approved by the local ethics committee of the Charité (approval no. EA½12/08).

Animal experimentation: All animal experiments were carried out in accordance to the national ethic principles (registration no. T0344/12, Charité).

## Additional files

### Supplementary files

• Supplementary file 1. List of primers and antibodies. (A) List of primers for qPCR and genotyping. (B) List of primary and secondary antibodies.

### Major datasets

The following dataset was generated:

| Author(s) | Year | Dataset title | Dataset URL | Database, license, and accessibility information |
|---|---|---|---|---|
| Hartmann B, Wai T, Hao Hu, MacVicar T, Musante L, Fischer-Zirnsak B, Stenzel W, Graef R, van den Heuvel L, Ropers HH, Wienker TF, Hübner C, Langer T, Kaindl AM | 2016 | Homozygous YME1L1 mutation causes mitochondriopathy with optic atrophy and mitochondrial network fragmentation | http://www.ncbi.nlm.nih.gov/gquery/?term= SRP073309 | Publicly available at the NCBI Gene Expression Omnibus (accession no: SRP073309) |

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
