## [Decision Letter]

Thank you for submitting your article "Homozygous *YME1L1* Mutation Causes Mitochondriopathy with Optic Atrophy and Mitochondrial Network Fragmentation" for consideration by *eLife*. Your article has been reviewed by two peer reviewers, and the evaluation has been overseen by a Reviewing Editor and Harry Dietz as the Senior Editor. The reviewers have opted to remain anonymous.

The reviewers have discussed the reviews with one another and the Reviewing Editor has drafted this decision to help you prepare a revised submission.

Summary:

In this manuscript, the authors identify a R149W genetic mutation in the mitochondrial inner membrane protease YME1L in patients with many neurologic phenotypes including intellectual disability, motor development delay, optic nerve atrophy, hearing impairment and variations in brain size. In addition, patients presented with other idiosyncratic phenotypes including other neurologic defects. The authors show that this mutation impairs maturation of YME1L by inhibiting processing of the precursor YME1L polypeptide by mitochondrial processing protease (MPP). This impaired processing destabilizes YME1L^R149W^ and reduces, but does not completely ablate, YME1L protein levels and activity. The authors then use overexpression to demonstrate that the degradation of precursor YME1L is mediated through an ATP-dependent autocatalytic mechanism. Furthermore, they demonstrate that YME1L^R149W^ impairs regulation of mitochondria morphology leading to fragmentation and decreases cell proliferation – both phenotypes observed in YME1L-deficient cells. However, the extent of these phenotypes in YME1L^R149W^ expressing patient fibroblasts is less than that observed in YME1L deficient cells, reflecting the residual activity of the mutant. Finally, the authors indicate that YME1L^R149W^-expressing fibroblasts have the same levels of ETC activity as controls, but have lower levels of mitochondria markers, suggesting a reduction in mitochondrial mass. Ultimately, these results demonstrate that reductions in YME1L activity afforded by the R149W YME1L mutation lead to neurologic defects in patients, demonstrating that YME1L activity is clinically important for the maintenance of mitochondrial biology in vivo.

Essential revisions:

Additional experiments are required to better characterize the residual YME1L activity observed in YME1L^R149W^-expressing patient fibroblasts. Does this result from a population of mature YME1L that accumulates in this cell or does it result from unprocessed YME1L^R149W^ that assembles into the active oligomer? Current data suggests the former, but more characterization of this active YME1L population in these cells is important to explain the more modest phenotypes observed in these cells as compared to YME1L-deficient cells.

The mechanism of YME1L^R149W^ degradation is not clear. It's not clear if YME1L^R149W^ is targeted to mitochondria efficiently in the first place or if it is degraded prior to entering mitochondria by other proteases (e.g., the proteasome). More experiments related to this point would be very helpful in further demonstrating exactly how this mutation influences YME1L activity in patient cells.

---

## [Author Response]

*Essential revisions:*

*Additional experiments are required to better characterize the residual YME1L activity observed in YME1L^R149W^-expressing patient fibroblasts. Does this result from a population of mature YME1L that accumulates in this cell or does it result from unprocessed YME1L^R149W^ that assembles into the active oligomer? Current data suggests the former, but more characterization of this active YME1L population in these cells is important to explain the more modest phenotypes observed in these cells as compared to YME1L-deficient cells.*

We have analyzed the ability of the unprocessed precursor form of YME1L1^R149W^ to assemble into the active oligomer by sucrose gradient centrifugation (Figure 3). To facilitate detection of the precursor form we have additionally introduced a mutation into the ATPase domain of YME1L1 to prevent autocatalytic degradation of YME1L1^R149W^ (Figure 3). Our results unambiguously demonstrate that the precursor form of YME1L1^R149W^ assembles into high molecular weight complexes. As we also observe low amounts of mature YME1L1^R149W^ in patient fibroblasts (Figure 1—figure supplement 1), the residual YME1L1^R149W^ activity in these cells may result either from this population or from assembled YME1L1^R149W^ precursor forms. We have now clearly stated these possibilities in the manuscript.

*The mechanism of YME1L^R149W^ degradation is not clear. It's not clear if YME1L^R149W^ is targeted to mitochondria efficiently in the first place or if it is degraded prior to entering mitochondria by other proteases (e.g., the proteasome). More experiments related to this point would be very helpful in further demonstrating exactly how this mutation influences YME1L activity in patient cells.*

We have directly examined mitochondrial targeting of YME1L1^R149W^ by immunofluorescence microscopy in *YME1L1*^-/-^ HeLa cells overexpressing YME1L1^R149W^. As shown now in Figure 2, the disease- associated mutation does not affect sorting of YME1L1 to mitochondria.

Moreover, we examined whether inhibition of the proteasome affects the accumulation of YME1L1^R149W^, but did not observe significant changes or stabilization in the levels of YME1L1^R149W^ in cells treated with the proteasomal inhibitor MG132 (Figure 2). These results indicate that YME1L1^R149W^ is not degraded by the proteasome prior to mitochondrial import. Rather, the mutant protein is degraded upon import into mitochondria in an autocatalytic fashion (Figure 3).